# Effects of Pretreatment on the Volatile Composition, Amino Acid, and Fatty Acid Content of Oat Bran

**DOI:** 10.3390/foods11193070

**Published:** 2022-10-03

**Authors:** Xue Bai, Meili Zhang, Yuanyuan Zhang, Yakun Zhang, Xinyue Guo, Rui Huo

**Affiliations:** College of Food Science and Engineering, Inner Mongolia Agricultural University, Huhhot 010018, China

**Keywords:** oat bran, volatile composition, amino acid, fatty acid

## Abstract

Pretreatment improves the edible quality of oat bran and prolongs the shelf life, whereas the effect of pretreatments (i.e., steaming(S-OB), microwaving(M-OB), and hot-air drying(HA-OB)) on the flavor characteristics of oat bran is unknown. This study identified volatile composition using HS-SPME/GC–MS and an electronic nose of oat bran. The amino acid compositions were determined by a High-Speed automatic amino acid analyzer and the fatty acids were determined by gas chromatography. The results showed that steaming and microwaving pretreatments enhanced the nutty notes of oat bran. Sixty-four volatile compounds in four oat brans were identified. OB exhibited higher aroma-active compounds, followed by S-OB, and M-OB, and the HA-OB had the lowest aroma-active compounds. Hexanal, nonanal, (E)-2-octenal,1-octen-3-ol, 2-ethylhexan-1-ol, and 2-pentylfuran were the key volatile compositions in oat bran. The aldehyde content decreased and the esters and ketones increased in steamed oat bran. Microwaving and hot air drying increased the aldehyde content and decreased the ester and alcohol content. Steamed oat bran had the lowest levels of total amino acids (33.54 g/100 g) and bitter taste amino acids (5.66 g/100 g). However, steaming caused a significant reduction in saturated fatty acid content (18.56%) and an increase in unsaturated fatty acid content (79.60%) of oat bran (*p* < 0.05). Hot air drying did not result in an improvement in aroma. The results indicated that steaming was an effective drying method to improve the flavor quality of oat bran.

## 1. Introduction

Oats are a healthy and nutritious food and a valuable livestock feed, world-renowned for their high protein content. In some regions, such as Inner Mongolia in China, oats are consumed as a staple food. The development of commercial food with oat as raw material has broad prospects [1]. Interest in the chemistry of oat flavor comes from consumer understanding of its health benefits [2]. A popular ingredient in cereals, muesli, breads, cookies, and snacks, oats are beloved for their pleasant nutty, brown, and grainy flavor [3]. Unlike other grain crops, oats are high in protein and lipids. It is well known that oats are rich in dietary fiber, especially water-soluble *β*-glucans. Oats also contain other minor elements, such as phenols [4]. Oat bran is the main by-product of oat processing, which has a high nutritional value, contains the highest proportion of β-glucan, and has significant biological activity [5,6,7]. Oat bran also contains bioactive substances such as polyphenols and the major triterpenoid saponins [8]. Oat bran products have grown in popularity due to growing demands for health and understanding of the physiological functions of bran. However, oat bran contains many enzymes. The lipolytic enzymes (mainly lipase) rapidly decompose the oil. This phenomenon causes oil rancidity and forms undesirable flavors, a peculiar smell, and a bitter taste [9]. Therefore, enzyme inactivation treatment is the premise of processing oat bran products.

Processing technology is the most important factor in flavor creation. Present pretreatment methods include steaming, microwaving, ohmic heating, infrared heating, extrusion puffing, and hot air drying [10]. Steam processing of oats inactivated enzymes and produced the unique oat flavor [11]. Ruge et al. observed remarkable differences in the flavor of oats after dry heat (hot air baking and infrared baking) and wet heat treatments (atmospheric pressure cooking, high-pressure cooking, microwaving). Compared to wet heat treatment, dry heat treatment improves the flavors of oats [12]. Huangfu Wenqian found that the types of volatile composition in Sino-Australian oat rice increased after steaming, and alkanes and aldehydes were the main flavor components of cooked oat groats [13]. Microwave heating has been used successfully as extraction support [14] to inactivate unwanted enzymatic activities such as the decomposition of *β*-glucan [15] or the decomposition of fat [16]. Joanna Harasym found that microwave heating of oatmeal for 3 min had the highest starch content and lowest GI, and it was the meal with highest carbohydrate equilibrium [17]. The preliminary results showed that the three pretreatment methods of steaming, microwaving, and oven heating improve antioxidant activity [18] and storage stability of the oat bran [19]. However, there are few technology applications for oat bran production and flavor enhancement.

The focus of flavor research is to determine the relationship between key volatile compounds and their sensory properties. A particular challenge in the study of the volatile composition of oats is the low and characteristic content of active flavor volatiles; thus, it lacks the sensitivity of the instrumental limit of detection required for identification [20]. Solid-phase microextraction (SPME) is a commonly used technique for the separation of volatile compounds that increases the sensitivity of dynamic headspace [21]. The use of SPME in oat analysis was first reported by Sides et al., as a comprehensive comparison of volatiles produced in the intermediate stages of processing from raw and de-grained oats to flaked oats [22]. The flavor of oats is mainly formed during the kinds of processing treatments, and the precursors and enzymes of natural oats are helpful in the formation of flavor. The aroma of processed oats is a complex combination of heat-related volatile aromas. Cooked oat flavors gradually become more oat-like, nutty, brown, or burnt During the heating of oats, two volatile components were found that make up the flavor of the oats: (1) Compounds produced by the oxidative decomposition of unsaturated fatty acids, and (2) Maillard reaction products, including heterocyclic pyrazine, pyrrole, and furan, and extruded oats produce roasted flavors. Heydanek and McGorrin found that typical Maillard-derived compounds such as pyrazine and furanones increased after heat treatment in oats [23,24]. Studies have shown that the astringency and bitterness of oats mainly come from avenamides and saponins [25]. This study aimed to adopt processing technologies (steaming, microwaving, and hot air drying) and evaluate their effects on flavor characteristics. We hypothesized that different treatments would lead to differences in the flavor profile of oat bran.

## 2. Materials and Methods 

### 2.1. Chemicals and Materials

The amino acid mixture standard solution (HPLC grade) was purchased from Wako Pure Chemical Industries, Ltd. (Shanghai, China). The fatty acid standards (HPLC grade) were purchased from Sigma-Aldrich (Shanghai, China). Hydrochloric acid (Guaranteed reagent,6 mol/L), sodium citrate(Guaranteed reagent), Methanol (HPLC grade), and sodium hydroxide (Guaranteed reagent,0.5 mol/L) were purchased from Sinopharm Chemical Reagent Co., Ltd. (Shanghai, China). 

### 2.2. Oat Bran Sample Preparation

The oat bran was obtained from Inner Mongolia Xibei Huitong Agricultural Technology Co., Ltd., Inner Mongolia, China. The oat bran processing parameters were chosen using previously published methods with minor modifications [18]. The oat bran was ground for 4 min at 50 Hz in a grinder (XL-20B, Guangzhou Xulang Machinery Equipment Co., Ltd. Guangzhou, China), then passed through a 0.420 mm sieve. The following are the thermal processing methods: 

Steaming oat bran (S-OB). The oat brans were steamed in a steamer at 100 °C for 20 min. (Midea Group Co., Ltd., Guangzhou, China). The completed process temperature was nearly 100 °C.

Microwaving oat bran (M-OB). The oat brans were microwaved for 2 min at 800 W (Guangdong Galanz Group Co., Ltd, Foshan, China). The completed process temperature was close to 100 °C.

Hot air drying (HA-OB). The oat brans were processed using the hot-air oven (Zhejiang Saide Instrument Equipment Co., Ltd., Shaoxing, China) at 140 °C for 15 min. The completed process temperature was 140 °C.

The oat bran samples that had been processed using the three different methods were cooled to room temperature on a cooling tray for approximately 40 min before being stored in a refrigerator at −20 °C. 

### 2.3. Analysis Methods of Oat Bran

#### 2.3.1. Color Analysis

The color of oat bran was measured based on a method previously reported by Kayitesi et al. with slight modifications [26]. The color of the samples was measured using a tristimulus colorimeter (CR-400 Chroma Meter, Konica Minolta Sensing, Osaka, Japan). The color was expressed in terms of lightness ( L*), red/green (a*), and blue/yellow characteristics (b*) after standardization with a white tile supplied by the manufacturer. For each batch of oat bran, three parallel samples were implemented.

#### 2.3.2. Electronic Nose Analysis

Electronic nose analysis of oat bran was based on previously reported methods with slight modifications [27]. Electronic nose measurements were performed using a PEN3 electronic nose analysis system (Airsense, Germany). The instrument was comprised of 10 metal oxide sensors and combined with a headspace sampler. Oat bran (2.50 g) was put into a 50 mL electronic nose injection bottle, immediately covered with teflon rubber, and equilibrated at 80°C for 30 min. The manual headspace sampling method was used for the detection, and the zero gas needle and the electronic nose sample needle were inserted into the sealed headspace tube for sampling at the same time. The parameters were set as follows: dry air was used at a flow rate of 400 mL/min as carrier gas. Washing time 60 s; zero setting time 5 s; sample preparation time 5 s; detection time 120 s.

#### 2.3.3. HS-SPME-GC-MS Analysis

The volatile compositions were analyzed following the methods of Jing-Nan et al. with a minor modification [28]. An SPME (Solid Phase Microextraction) manual device equipped with 50 μm/30 μm divinylbenzene/carboxy/polydimethylsiloxane (DVB/CAR/PDMS) fibers (Supelco, Bellefonte, PA, USA) was used to extract volatiles from the sample sexual compounds. Oat bran homogenate (2.0 g oat bran powder in 20 mL of saturated NaCl solution) was added to a 50 mL vial equipped with a magnetic stir bar. The samples were equilibrated at 80 °C for 30 min before fibers were inserted into vials and volatile compounds were extracted for 1 h. Finally, the fiber was inserted into the injection port of the GC and desorbed in the splitless mode for 5 min. Analyses were performed in a TRACE™ 1300 gas chromatography system (Thermo Fisher Scientific, Waltham, MA, USA). A chromatographic column, TG–5 MS (30 m *×* 0.25 mm *×* 0.25 μm ), was used. Chromatographic conditions were set as follows: the carrier gas was helium with a flow rate of 1 mL/min; the injector temperature was 250 °C; the oven temperature programming was set at 100°C for 5 min, then 8 °C/min to 200 °C, held for 5 min, finally 15°C/min to 280°C, maintained for 15 min. Helium (99.999%, AGA-Fano) was used as carrier gas, with linear velocity at a constant flow (1 mL/min). The ionization source temperature was set at 280 °C. Mass spectra and reconstructed (total) ion chromatograms were obtained by scanning in the mass range m/z 40–600 Da. Chromatographic peaks were checked for homogeneity with the aid of the mass chromatograms for the characteristic fragment ions and with the help of the peak purity program. Tentative identification criteria of compounds were based on coincidence percentage (≥80%) of obtained compounds compared to NIST14 mass spectra libraries. The relative content of each component was calculated by the peak area normalization method.

#### 2.3.4. Analysis of Amino Acids Content

The amino acid contents were analyzed following the method of Huang, M. et al. with a minor modification [29]. Around 300 mg of oat bran powder was weighed and placed in a digestive tube. Next, 10 mL of 6 mol/L HCl solution was added and ultrasonicated for 2 min. The tube was sealed with nitrogen purging. After being cooled, the digestion tube was shaken at room temperature and its cap was opened. The solution was diluted with ultrapure water in a 100 mL volumetric flask and mixed thoroughly. It was then filtered through a 0.45 µm inorganic filter membrane, and 2.5 mL of this filtrate was transferred into a 25 mL volumetric flask and diluted with ultrapure water. This solution was again filtered using a 0.45 µm inorganic filter membrane. This filtrate was used for detection and analysis on a LA8080 High-Speed automatic amino acid analyzer (Hitachi High-Tech Science Corporation, Japan). 

#### 2.3.5. Analysis of Fatty Acid Composition

The fatty acid contents were analyzed following the method of Khan, N.A et al. with a minor modification [30]. The crude fat in oat bran was extracted with petroleum ether. Two milliliters of 2% sodium hydroxide methanol solution was added to oat bran oil in a water bath at 85°C for 30 min. Next, 3 mL of boron trifluoride methanol solution (14%) was added to the boiling solution from the top of the condenser with a pipette and allowed to boil for 30 min. Subsequently, 5 mL of isooctane was added to the boiling mixture from the top of the condenser. The heating was stopped immediately, the condenser was removed, and the flask was taken out. Then, 20 mL of saturated sodium chloride solution was added, and the flask was closed with a stopper and shaken vigorously for at least 15 s. The saturated sodium chloride solution was added continuously to the neck of the flask and allowed to stand for stratification. Furthermore, the upper layer solution (isooctane layer) was transferred into a test tube using a pipette. An appropriate amount of anhydrous sodium sulfate was added to remove trace water from the solution. Finally, 2 uL of this solution was directly injected into the 7890A gas chromatography system (Agilent Technologies Inc, USA) for analysis. The gas chromatography conditions were as follows: column: CD-2560, 100 m × 0.25 mm × 0.24 µm; vaporization chamber temperature: 250 °C; detector: hydrogen flame ionization detector (FID) temperature: 250 °C; column temperature: 210 °C; carrier gas: helium with a flow rate of 0.5 mL/min; oven temperature programming: set at 130°C for 5 min, then 4 °C/min to 240 °C, held for 30 min.; injection method: manual injection, split less; pre-column pressure: air 55 kPa, hydrogen 65 kPa, nitrogen 80 kPa; carrier gas: high-purity N2; gas: H2; auxiliary gas: air.

#### 2.3.6. Statistical Analysis

The results were expressed as mean values ± standard deviation (*n* = 3). Analysis of variance and Duncan test with *p* < 0.05 were carried out using the SPSS 26 (IBM, Armonk, NY, USA). The PCA and heatmap were performed using Origin 2018 statistical software (OriginLab, Northampton, MA, USA). The input variables for PCA were the analysis of electronic nose volatile compounds in four kinds of oat bran. PCA can summarize the raw information of this variable and compress the cube. The original correlated variable can be transformed into several independent uncorrelated variables that explain most of the variability in the original variable. Correlation analysis was performed using Spearman analysis. Spearman rank correlation coefficient is a nonparametric statistical method, which is suitable for data with non-normal distribution. The correlation coefficient rho (R value) is the value between −1 and 1. When −1 < rho < 0, they are negatively correlated; when 0 < rho < 1, they are positively correlated; when rho = 0, there is no correlation between them.

## 3. Results and Discussion

### 3.1. Color Analysis of Pretreated Oat Bran

Color analysis (L*, a*, and b* values) were studied on each oat bran, which were the primary indicator of individual evaluation. As can be seen from Table 1, HA-OB has the highest L* value of 80.24, M-OB is 78.85, S-OB is 75.16, and OB is 75.01. This may be related to the browning of oat bran at the heating temperature. When the degree of browning and non-enzymatic caramelization is higher, the caramel color is darker [31]. In the initial stage of heat processing, Maillard reactions and caramelization reactions can produce colored compounds [32]. Moreover, S-OB exhibited high a* and b* values compared with OB. Color has been shown to correlate with roasted aroma and is often used as a measure to judge the degree of heating [33]. Therefore, the color of steamed oat bran has certain advantages over other methods.

### 3.2. Electronic Nose Analysis of Oat Bran Samples

By imitating the human olfactory system, the electronic nose can obtain comprehensive information about taste in a short time [34]. In order to further study the effect of different pretreatment methods on the overall flavor of oat bran, the electronic nose was used to evaluate the flavor of oat bran. Principal component analysis (PCA) is able to provide better visualization and highlight differences in volatile characteristics. PCA is a multivariate stoichiometric technique that can be used to identify patterns of correlation between component variables that lead to differences between samples. Figure 1 shows the results of the principal component analysis (PCA) of electronic nose in oat bran.. The values were 96.74% (PCA1) and 2.32% (PCA2), and the cumulative differentiation index was 99%. The results showed that PCA could better reflect the general odor characteristics of oat bran with different pretreatment methods. Although the sensor was unstable during replication, volatile composition from different pretreated oat brans were located in four distinct regions. Figure 1 shows that the odor response distributions of the S-OB, M-OB, and HA-OB samples were far from the OB samples, which indicated that steaming, microwaving, and hot air drying changed the odor of oat bran. Moreover, the S-OB and M-OB samples were closer, indicating that steaming and microwaving processes had similar aroma. In conclusion, the three heat treatments could change the oat bran aroma profiles.

### 3.3. HS-SPME-GC-MS Analysis of Volatile Compounds of Oat Bran

To further study the effect of different heating methods on the volatile composition of the oat bran, we used HS-SPME-GC-MS to determine the specific volatile components of the oat bran samples. The above chromatograms were retrieved by the NIST14 spectral library and the manual map analysis. The volatile compounds of the fresh and heated oat bran samples are shown in Table 2. A total of 64 volatile compounds from oat bran were tentatively identified and quantified, including 14 aldehydes, 12 esters, 10 alcohols, 5 ketones, 5 acids, 4 amines, 11 alkanes, 2 pyrazines, and 1 furan. A total of 43, 36, and 36 compounds were detected in OB, S-OB, and M-OB samples, respectively. It was noted that only 25 compounds were detected in HAOB. Among the 64 aroma compounds detected in oat bran, hexanal, nonanal, (E)-2-octenal, 1-octen-3-ol, 2-ethylhexan-1-ol, and 2-pentylfuran contributed the most. These compounds mainly contributed to the “fresh”, “green”, “waxy”, “aldehydic”, “earthy”, and “fruity” characteristics. Heydanek and McGorrin [23] initially reported 111 volatiles in oat groats, hexanal was the most significant, with an odor threshold of 4.5 ng g−1, which was similar to the results of this experimental study. The high temperature could produce a variety of volatile flavor components by the Maillard reaction, lipid oxidation reaction, and amino acid degradation reaction [35].

Aldehydes generally have the smell of cream, fat, vanilla, and fragrance. The aldehydes produced by the three heat treatment methods were absolutely dominant in relative content and quantity, and the thresholds of aldehydes were generally low, which greatly contributed to the overall aroma of oat bran and were the main volatile components of oat bran. Aldehydes accounted for 59.01%, 51.68%, 64.76%, and 68.13% of the volatile components in OB, S-OB, M-OB, and HA-OB, respectively. The aldehyde content of the oat bran increased with microwaving and hot air drying. This may be due to aldehydes derived primarily from the oxidation and degradation of unsaturated fatty acids [36]. In addition, increasing the temperature of oat bran samples may also promote the Maillard reaction, which increases aldehyde content [37]. Hexanal, heptanal, nonanal, (E)-hept-2-enal, and (E,E)-2,4-decadienall mainly provided a grassy aroma. The content of hexanal in the HA-OB samples was highest (54.38%), followed by the M-OB content of 49.31%, and the S-OB was the lowest (36.49%). Similar results have been reported in other oats [20,23]. The highest concentration volatile in oat bran is hexanal. The concentration of hexanal increases after heating. The contents of nonanal and (E)-2-octenal in the S-OB and M-OB were higher than that in other samples. Nonanal presents waxy, citrus scents. The aroma produced by the aldehydes gives the oat bran its unique fragrance.

Ten alcohols were detected in oat bran, including heptan-1-ol and 1-octen-3-ol, which were all identified in four samples. The content of 2-ethylhexan-1-ol in S-OB was higher than that in OB and M-OB. It was not detected in HA-OB. The odor of 1-octen-3-ol is often described as mushroom-like [38]. Aldehydes and alcohols played an important role in the flavor profiles of oat bran, which were consistent with previous research. Two other studies investigated the effect of heat treatment on the formation of volatile oats at various stages, from raw oat,, hulled, and dry hulled, to oat flakes [39,40]. As shown in Table 1, OB had the highest amount of alcohols (17.94%), followed by the S-OB (15.10%) and M-OB(12.22%), the HA-OB had the lowest content (5.96%), which indicated that the content of alcohols decreases with higher heating temperature.

The content of esters in oat bran with different treatments was less (<10%). Among the 12 kinds of esters detected, S-OB had the highest content (7.56%), followed by the OB (5.25%), which may be related to the thermal degradation and oxidation of fat or fatty acid. Zhou et al. studied that the majority of oat volatile compositions were produced by heating and cooked oat porridge showed more volatiles [41]. The OB contained two kinds of ketones (1.82%), S-OB produced five kinds of ketones (5.34%), mainly including (2,5-dioxopyrrolidin-1-yl) benzoate(1.70%), and (E)-oct-3-en-2-one(1.40%), M-OB produced three kinds of ketones(2.97%), and HA-OB one kind of ketone (0.63%). Alcohols, aldehydes, and ketones have been reported to play an important role in the flavor profiles of foods [42,43]. It has been reported that ketones are mainly obtained by the Maillard reaction and oxidative degradation of unsaturated fatty acids [44].

The content of typical Maillard reaction products, such as pyrazine and furan, increases after heat treatment. The S-OB and M-OB contained two kinds of pyrazines, whereas the HA-OB did not contain pyrazines. The content of 2-pentylfuran in S-OB, M-OB, and HA-OB was higher than that in OB. This 2-Pentylfuran is the oxidation product of linoleic acid. Those substances had the lowest odor thresholds, which suggested that compounds, such as 2,5-dimethylpyrazine, 2-ethyl-3,5-dimethylpyrazine, and 2-pentylfuran, as the degradation reaction products of the Strecker reaction, were important to the nutty flavor developed in pretreated oat bran. Pyrazines have a toast or nut-like taste. They are formed by the interaction of amino acids and carbohydrates, with the highest proportion of the three types of heterocycles.

The flavor of oat bran was mainly formed during heat treatment, and the precursors and enzymes in natural oat bran were helpful for flavor formation. Pretreated oat bran was progressively more oat-like and nutty. Two classes of volatile components were detected in heated oats: (1) compounds derived from the oxidative decomposition of unsaturated fatty acids, resulting in saturated and unsaturated aldehydes and ketones, (2) Maillard reaction products, including heterocyclic pyrazine, pyrrole, furan, and sulfur-containing compounds, which are rich in high temperature, low moisture, extruded oats that produce toasted, roasted flavors [20]. Heydanek and McGorrin found that typical Maillard-derived compounds such as pyrazine and furanones were increased after heat treatment in oat [23,24]. Compared with microwaving and hot air drying, steamed oat bran has a better flavor, probably because the steaming treatment temperature is lower, but the heating time is longer (100 °C, 20 min). This processing method results in a higher moisture content of the sample. The starch in the oat bran expands and gelatinizes with water absorption, partially affecting the flavor [45]. 

### 3.4. Effects of Heating Methods on Amino Acids of Oat Bran

#### 3.4.1. Differences in the Contents and Composition of Amino Acids in Oat Bran

From Table 3, the content of total free amino acids in fresh and heat-treated oat bran ranged from 17.40 to 18.66 g/100 g dry weight. Oat is rich in lysine, which is lacking in other grains. The lysine content of oat bran did not change significantly after the three pretreatment methods (*p* > 0.05). Among the branched-chain amino acids, isoleucine decreased significantly in the S-OB, M-OB, and HA-OB groups, valine displayed no significant change, and leucine decreased significantly in HA-OB (*p* < 0.05). Compared to the oat bran, the content of essential amino acids and total amino acids was reduced after steaming, microwaving, and hot air drying, but the change was insignificant (*p* > 0.05). The content of total amino acids in HA-OB samples was the lowest. This may be due to the high temperature that promotes protein hydrolysis. Ala, Gly, Ser, Thr, and Pro gave a sweet taste; Glu and Asp gave pleasant umami; Val, Met, Leu, Ile, Phe, His, and Arg gave a bitter taste. Compared with OB, S-OB, M-OB, and HA-OB gave lower contents of sweet amino acids, umami amino acids, and bitter amino acids, indicating that the three treatments could reduce the non-volatile substances (bitter, rancidity, etc.) and retain the content of volatile composition (specific aroma substances) in oat bran. It was studied that umami and sweet amino acids have higher activity [46]. In addition, in oat bran, sweet, umami, and bitter amino acid flavors dominate the overall flavor. S-OB and M-OB enhance the flavor of oat bran. As revealed in Figure 2, fresh oat bran contained the sweetest and most umami-like amino acids, while the S-OB sample contained the lowest bitter tasting amino acids.

#### 3.4.2. Analysis of the Correlation of Amino Acids and Flavor Components in the Pretreated Oat Bran

The Spearman correlation coefficient is a nonparametric index to measure the dependence of two variables. It uses a monotone equation to evaluate the correlation between two statistical variables. If there are no duplicate values in the data, and when the two variables are completely monotonically correlated, the correlation coefficient rho (R value) is the value between −1 and 1. Red represents positive correlation, blue represents negative correlation. As shown in Figure 3 and Table 2, most of the amino acids, except Pro, Met, and Tyr, displayed an extremely significant positive correlation with components: VFC42, VFC34, VFC43, and VFC26(*p* < 0.01). These components were identified as acetic acid, 4-ethylcyclohexan-1-ol, octanoic acid, and decanoic acid ethyl ester. The umami amino acid glutamic acid (Glu) displayed an extremely significant positive correlation with components VFC17, VFC38, VFC55, VFC53, VFC33, VFC35, VFC36, and VFC48, and a very significant negative correlation with components VFC5, VFC4, VFC22, and VFC2 (*p* < 0.01). These components were identified as aldehydes, esters, alcohols, and alkanes. The bitter amino acid leucine acid (Leu) showed an extremely significant positive correlation with components VFC17, VFC38, VFC55, VFC53, VFC33, VFC35, VFC36, VFC48, VFC62, and VFC12, and a very significant negative correlation with the components VFC5, VFC4, VFC7, and VFC51 (*p* < 0.01). Most of those components were identified as aldehydes, esters, alcohols, and alkanes. The above comprehensive analysis demonstrated that most amino acids were mainly related to alcohols, esters, and aldehydes. Amino acids are the precursors in the metabolic pathway of volatile composition. Since heat treatment aggravates the Maillard and Strecker degradation reactions, the relationship between aromatic ester substances and amino acids is mainly due to amino acid metabolism. Transamination and decarboxylation occur. Hence, the changes in the free amino acids are closely related to the metabolism of volatile flavor substances [20].

### 3.5. Effects of Heating Methods on Fatty Acids of Oat Bran

#### 3.5.1. Differences in the Contents and Composition of Fatty Acids in Oat Bran

Oleic and linoleic acids are the primary fatty acids in oats [47]. The saturated fatty acid content in oat bran was 18.88%, and palmitic acid contributed the highest amount (Table 4). The content of unsaturated fatty acid was 79.32%, wherein the proportion of oleic acid (46.98%) and linoleic acid (31.68%) was higher. Compared to the OB group, the saturated fatty acid content in the S-OB group had reduced significantly, while the unsaturated fatty acid content had increased significantly (*p* < 0.05). Further, the saturated fatty acid and unsaturated fatty acid content of the M-OB and HA-OB groups had decreased (*p* > 0.05). This observation was consistent with an increase in the aldehyde content in the microwaved and hot air dried oat bran, mainly due to the oxidation and degradation of unsaturated fatty acids. Compared to microwaving and hot air drying, steam treatment increased the amount of unsaturated fatty acids in oat bran, which is highly beneficial to maintaining human health.

#### 3.5.2. Correlation Analysis of Fatty Acids and Flavor Components in the Pretreated Oat Bran

The correlation analysis demonstrated that linoleic acid (C18:2) displayed a significant positive correlation with VFC23, VFC45, VFC19, and VFC40 (*p* < 0.05), whereas oleic acid (C18:1) and palmitic acid (C16:0) displayed a significant negative correlation with those substances (Figure 4, Table 2) (*p* < 0.01). The constituents were detected only in S-OB. Oleic acid (C18:1) exhibited an extremely significant positive correlation with VFC13, VFC28, VFC33, VFC35, VFC42, VFC34, VFC43, and VFC26 (*p* < 0.01). The constituents were (E, E)-2,4-decadienal, 1-octen-3-ol, (Z)-4,5-dimethylhept-2-en-3-ol,4,4,6-trimethyl-cyclohex-2-en-1-ol, acetic acid, 4-ethylcyclohexan-1-ol, octanoic acid, and decanoic acid ethyl ester, respectively. Linoleic acid (C18:2) exhibited an extremely significant positive correlation with VFC31, VFC56, VFC32, VFC39, and VFC14 (*p* < 0.01). The components comprised mainly aldehydes and alcohols. The products of lipoxygenase detected in oats were 2-Pentylfuran, 1-octen-3-ol, (E)-hept-2-enal, hexanal, and 2-heptanone. However, the action of oat lipoxygenase on linoleic acid produces mostly hexanal and 2-nonenal. Different processing methods alter the release of volatile compounds by affecting the correlation between factors involved in amino acid and fatty acid metabolism.

## 4. Conclusions

The effects of three different pretreatment methods (S-OB, M-OB, and HA-OB) on the flavor properties of oat bran were investigated. Compared with OB, the S-OB color had a more obvious advantage than the other methods and could also improve the bitter flavor. S-OB and M-OB samples could enhance the release of flavor components from oat bran, producing a stronger degree of taste. HA-OB could enhance the fresh and green aroma. Steaming, microwaving, and hot air drying had no significant effect on the content and types of essential amino acids and total amino acids in oat bran (*p* > 0.05). Steaming caused a significant reduction in the saturated fatty acid content and an increase in the unsaturated fatty acid content (*p* < 0.05). Most amino acids and fatty acids were mainly related to alcohols, esters, and aldehydes. The growth in daily consumption of oats and the oat bran market is driven by health benefits, such as reducing the risk of developing different chronic diseases. The development of oat bran is greatly limited due to its high oil content and easy rancidity. In order to extend shelf life and provide nutrient dense and high-quality products to consumers, processing research should be conducted to better understand the mechanisms of nutrient and flavor formation changes. The research results provide up-to-date information about the volatile compounds that contribute to the flavor of oat brans during processing, which could provide useful information to the oat bran industry to select the optimum processing method. The specific mechanism of action needs further study in combination with changes in the enzymes.

## Figures and Tables

**Figure 1 foods-11-03070-f001:**
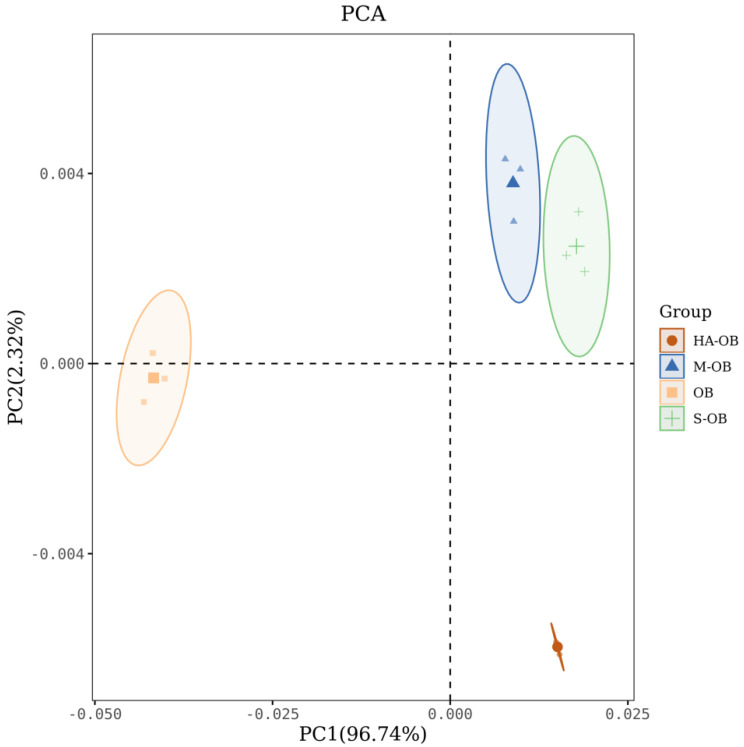
PCA scatter plot of electronic nose in different pretreatments. OB, Oat bran; S-OB, Steamed oat bran; M-OB, Microwaved oat bran; HA-OB, Hot air dried oat bran.

**Figure 2 foods-11-03070-f002:**
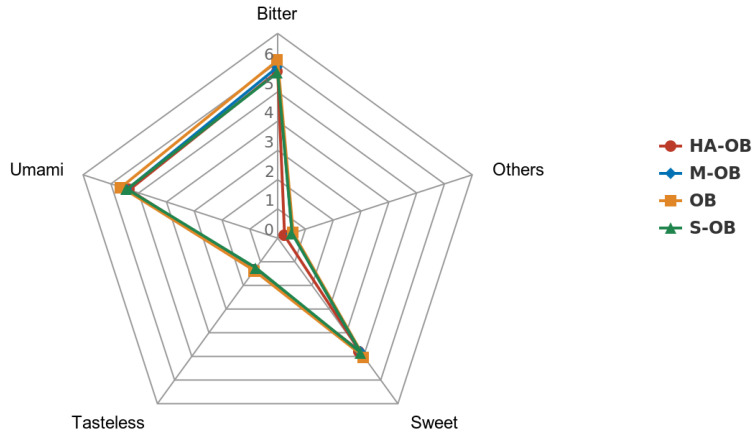
Radar graph of the sensory results of pretreated oat bran. OB, Oat bran; S-OB, Steamed oat bran; M-OB, Microwaved oat bran; HA-OB, Hot air dried oat bran. Sweet = Ala + Gly + Ser + Thr + Pro; Umami = GLU + ASP; Bitter =Val + Met + Leu + Ile + Phe + His + Arg; Tasteless = Lys + Tyr; Other = Cys.

**Figure 3 foods-11-03070-f003:**
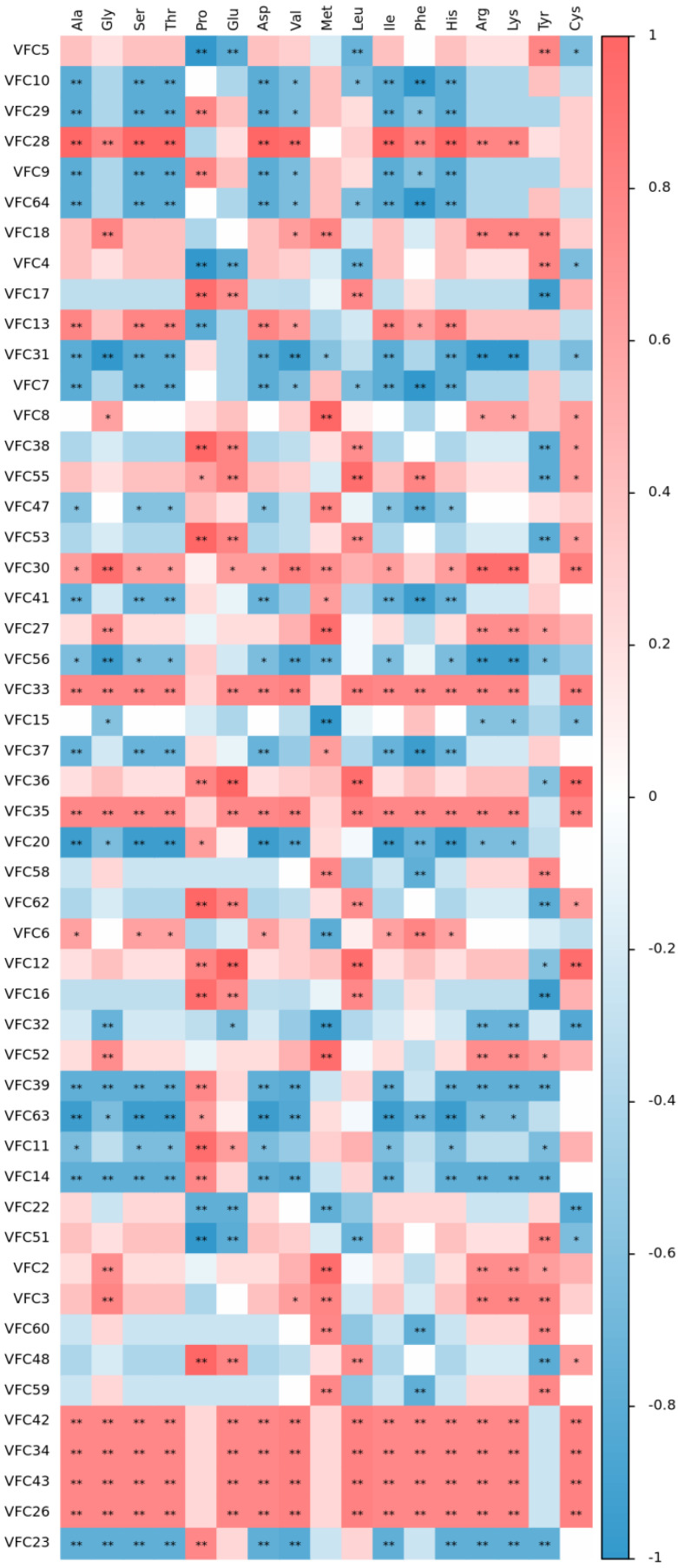
Correlation heat map between amino acids and flavor substances in pretreated oat bran. Red represents positive correlation, blue represents negative correlation, * represents *p* < 0.05, ** represents *p* < 0.01.

**Figure 4 foods-11-03070-f004:**
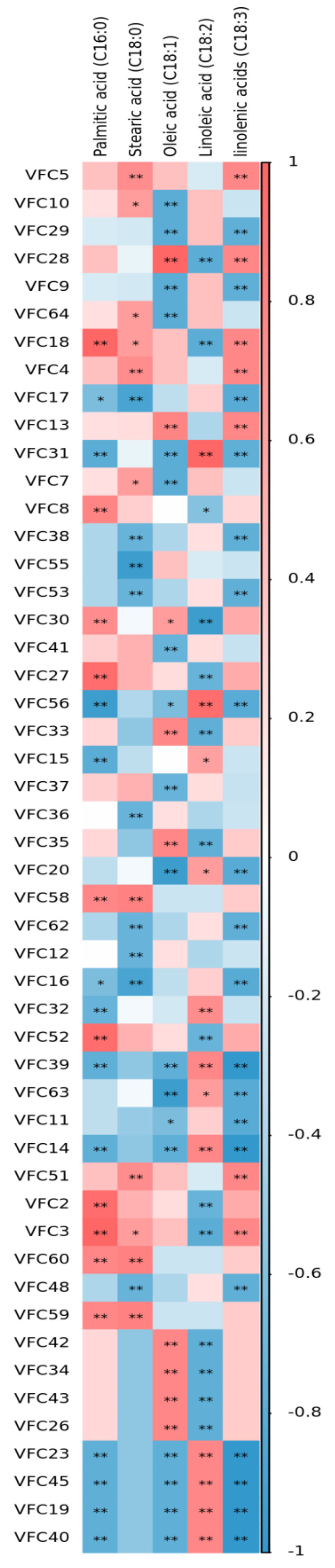
Correlation heat map between fatty acids and flavor substances in pretreated oat bran. Red represents positive correlation, blue represents negative correlation, * represents *p* < 0.05, ** represents *p* < 0.01.

**Table 1 foods-11-03070-t001:** Color analysis of the different treated oat bran.

	OB	S-OB	M-OB	HA-OB
L* value	75.01 ± 0.37 c	75.16 ± 0.29 c	78.85 ± 0.13 b	80.24 ± 0.53 a
a* value	3.45 ± 0.08 b	3.92 ± 0.12 a	3.21 ± 0.04 c	2.80 ± 0.04 d
b* value	15.01 ± 0.17 b	15.95 ± 0.21 a	14.87 ± 0.13 b	14.38 ± 0.13 c

Mean ± SD (*n* = 3). Means with different superscript (a,b,c,d) in the same row are significantly different (*p* < 0.05).OB, Oat bran; S-OB, Steamed oat bran; M-OB, Microwaved oat bran; HA-OB, Hot air dried oat bran.

**Table 2 foods-11-03070-t002:** Composition and relative content of volatile composition of oat bran under different pretreatments.

Serial Number	Volatile Compound	Retention Time (min)	CAS	Molecular Formula	Relative Content/(%)			
					OB	S-OB	M-OB	HA-OB
Aldehydes(14)								
VFC1	Butanal	1.27	123-72-8	C4H8O	ND	ND	0.14 ± 0.01	ND
VFC2	3-Methylbutanal	1.65	590-86-3	C5H10O	0.45 ± 0.02	ND	0.50 ± 0.01	ND
VFC3	2-Methylbutanal	1.69	96-17-3	C5H10O	0.30 ± 0.01	ND	0.32 ± 0.02	0.22 ± 0.01
VFC4	Pentanal	1.93	110-62-3	C5H10O	1.33 ± 0.31	1.14 ± 0.02	1.55 ± 0.14	1.64 ± 0.14
VFC5	Hexanal	3.14	66-25-1	C6H12O	44.98 ± 3.25	36.49 ± 2.98	49.31 ± 2.15	54.38 ± 1.98
VFC6	(E)-Hex-2-enal	4.14	6728-26-3	C6H10O	0.47 ± 0.02	0.41 ± 0.02	0.33 ± 0.01	0.64 ± 0.01
VFC7	Heptanal	5.22	111-71-7	C7H14O	0.97 ± 0.03	1.24 ± 0.14	1.30 ± 0.21	1.15 ± 0.02
VFC8	(E)-Hept-2-enal	6.63	57266-86-1	C7H12O	1.31 ± 0.0.1	1.05 ± 0.02	1.36 ± 0.05	ND
VFC9	(E)-2-Octenal	9.48	2548-87-0	C8H14O	2.37 ± 0.02	3.22 ± 0.17	2.52 ± 0.10	1.83 ± 0.13
VFC10	Nonanal	10.85	124-19-6	C9H18O	3.80 ± 0.03	5.76 ± 0.21	5.96 ± 0.18	5.43 ± 0.11
VFC11	Decanal	13.77	112-31-2	C10H20O	0.28 ± 0.01	0.52 ± 0.01	0.28 ± 0.01	0.24 ± 0.01
VFC12	(2E,4E)-Nona-2,4-dienal	13.96	5910-87-2	C9H14O	0.64 ± 0.01	0.59 ± 0.01	0.54 ± 0.02	ND
VFC13	(E,E)-2,4-Decadienal	16.78	25152-84-5	C10H16O	2.11 ± 0.09	ND	0.79 ± 0.01	2.60 ± 0.08
VFC14	4-Dodecoxybenzaldehyde	18.98	24083-19-0	C19H30O2	ND	1.26 ± 0.06	ND	ND
	Total				59.01 ± 3.80	51.68 ± 3.64	64.76 ± 2.91	68.13 ± 2.49
Esters(12)								
VFC15	Ethenyl hexanoate	7.4	3050-69-9	C8H14O2	0.31 ± 0.01	0.68 ± 0.05	ND	1.69 ± 0.13
VFC16	Benzyl N-aminocarbamate	9.05	5331-43-1	C8H10N2O2	0.68 ± 0.02	1.08 ± 0.04	ND	ND
VFC17	Phosphonofluoridic acid, methyl-, octyl ester	9.54	144313-52-0	C9H20FO2P	1.74 ± 0.11	3.88 ± 0.18	ND	ND
VFC18	Acetic acid, 2-ethylhexyl ester	12.21	103-09-3	C10H20O2	1.63 ± 0.15	ND	2.66 ± 0.19	1.45 ± 0.15
VFC19	Acetoxyacetic acid, nonyl ester	12.79	1000308-31-4	C13H24O4	ND	0.33 ± 0.05	ND	ND
VFC20	Formic acid, 2-ethylhexyl ester	14.71	1000368-94-7	C9H18O2	ND	1.25 ± 0.09	0.80 ± 0.02	ND
VFC21	Acetic acid, 2-phenylethyl ester	15.19	103-45-7	C10H12O2	0.22 ± 0.01	ND	ND	ND
VFC22	Oxalic acid, allyl pentadecyl ester	17.99	1000309-24-3	C20H36O4	ND	ND	ND	1.26 ± 0.14
VFC23	Propanoic acid, 3-chloro-, 4-formylphenyl ester	19.98	1000142-41-5	C10H9ClO3	ND	0.34 ± 0.01	ND	ND
VFC24	Phenol, 2,6-bis(1,1-dimethylethyl)-4-methyl-, methylcarbamate	21.82	6881	C17H27NO2	0.16 ± 0.02	ND	ND	ND
VFC25	Oxalic acid, allyl pentadecyl ester	25.08	1000309-24-3	C20H36O4	0.16 ± 0.03	ND	ND	ND
VFC26	Decanoic acid, ethyl ester	33.81	110-38-3	C12H24O2	0.35 ± 0.01	ND	ND	ND
	Total				5.25 ± 0.36	7.56 ± 0.42	3.46 ± 0.21	4.40 ± 0.42
Alcohols(10)								
VFC27	Heptan-1-ol	7.02	111-70-6	C7H16O	0.72 ± 0.02	0.68 ± 0.04	0.95 ± 0.14	0.68 ± 0.02
VFC28	1-Octen-3-ol	7.27	3391-86-4	C8H16O	3.42 ± 0.14	2.06 ± 0.11	2.17 ± 0.32	2.64 ± 0.11
VFC29	2-Ethylhexan-1-ol	8.67	104-76-7	C8H18O	5.23 ± 0.22	9.04 ± 0.24	5.96 ± 0.41	ND
VFC30	Octa-3,5-dien-2-ol	8.95	69668-82-2	C8H14O	1.8 ± 0.16	ND	1.33 ± 0.05	ND
VFC31	Octan-1-ol	9.87	111-87-5	C8H18O	ND	2.01 ± 0.05	1.35 ± 0.06	1.50 ± 0.09
VFC32	3,5-Dimethylcyclohexan-1-ol	10.53	5441-52-1	C8H16O	ND	0.74 ± 0.01	ND	0.86 ± 0.02
VFC33	(Z)-4,5-Dimethylhept-2-en-3-ol	10.53	55956-37-1	C9H18O	2.90 ± 0.14	ND	ND	ND
VFC34	4-Ethylcyclohexan-1-ol	16.2	4534-74-1	C8H16O	0.47 ± 0.03	ND	ND	ND
VFC35	4,4,6-Trimethyl-cyclohex-2-en-1-ol	17.47	1000144-64-7	C9H16O	2.12 ± 0.08	ND	ND	ND
VFC36	2,4,7,9-Tetramethyldec-5-yne-4,7-diol	19.43	126-86-3	C14H26O2	1.28 ± 0.07	0.57 ± 0.03	0.46 ± 0.02	0.28 ± 0.01
	Total				17.94 ± 0.86	15.10 ± 0.48	12.22 ± 1.00	5.96 ± 0.25
Ketones (5)								
VFC37	5-Methylhexan-2-one	4.95	110-12-3	C7H14O	0.63 ± 0.02	0.65 ± 0.01	0.69 ± 0.05	0.63 ± 0.03
VFC38	(2,5-Dioxopyrrolidin-1-yl) benzoate	6.7	23405-15-4	C11H9NO4	1.19 ± 0.11	1.70 ± 0.05	0.56 ± 0.01	ND
VFC39	(E)-Oct-3-en-2-one	8.95	1669-44-9	C8H14O	ND	1.40 ± 0.08	ND	ND
VFC40	1-(2-Hydroxy-5-methylphenyl)ethanone	16.71	1450-72-2	C9H10O2	ND	0.26 ± 0.01	ND	ND
VFC41	2-Pentylcyclopentanone	17.44	1000191-05-3	C10H18O	ND	1.33 ± 0.13	1.72 ± 0.06	ND
	Total				1.82 ± 0.13	5.34 ± 0.28	2.97 ± 0.12	0.63 ± 0.03
Acids(5)								
VFC42	Acetic acid	1.31	64-19-7	C2H4O2	0.53 ± 0.01	ND	ND	ND
VFC43	Octanoic acid	12.79	124-07-2	C8H16O2	0.44 ± 0.02	ND	ND	ND
VFC44	Nonanoic acid	15.48	112-05-0	C9H18O2	0.13 ± 0.01	ND	ND	ND
VFC45	n-Decanoic acid	32.55	334-48-5	C10H20O2	ND	0.34 ± 0.02	ND	ND
VFC46	Pyridine-2-carboxylic acid	2.5	98-98-6	C6H5NO2	ND	ND	0.25 ± 0.01	ND
	Total				1.10 ± 0.04	0.34 ± 0.02	0.25 ± 0.01	0
Amines(4)								
VFC47	(1R)-1-Cyclohexylethanamine	1.02	5913-13-3	C8H17N	0.82 ± 0.02	1.16 ± 0.07	1.21 ± 0.06	ND
VFC48	2-Hydroxypropanamide	1.11	2043-43-8	C3H7NO2	0.22 ± 0.01	0.43 ± 0.02	0.11 ± 0.01	ND
VFC49	N-methylmethanamine	1.09	124-40-3	C2H7N	ND	ND	ND	0.15 ± 0.05
VFC50	1-Methoxypropan-2-amine	1.14	37143-54-7	C4H11NO	ND	ND	ND	0.30 ± 0.01
	Total				1.04 ± 0.03	1.59 ± 0.09	1.32 ± 0.07	0.45 ± 0.06
Alkanes(11)								
VFC51	2-Pentyloxirane	5.32	5063-65-0	C7H14O	0.33 ± 0.01	ND	0.36 ± 0.02	0.45 ± 0.02
VFC52	Dodecane	13.61	112-40-3	C12H26	0.67 ± 0.02	ND	0.86 ± 0.08	ND
VFC53	Tridecane	16.37	629-50-5	C13H28	0.80 ± 0.02	1.83 ± 0.14	0.53 ± 0.01	ND
VFC54	Hexadecane	21.48	544-76-3	C16H34	0.16 ± 0.01	ND	ND	ND
VFC55	2-Phenoxyethoxybenzene	28.3	104-66-5	C14H14O2	1.70 ± 0.08	1.25 ± 0.11	0.79 ± 0.02	0.31 ± 0.01
VFC56	2,4,6-Trimethyloctane	13.6	62016-37-9	C11H24	ND	2.47 ± 0.15	ND	0.48 ± 0.02
VFC57	2,3-Dimethyloxirane	1.36	3266-23-7	C4H8O	ND	ND	0.24 ± 0.01	ND
VFC58	Pentylcyclopropane	9.87	2511-91-3	C8H16	ND	ND	2.00 ± 0.14	ND
VFC59	3,7-Dimethyldecane	18.98	17312-54-8	C12H26	ND	ND	0.55 ± 0.01	ND
VFC60	2-(Oxiran-2-ylmethyl)-3-[3-(oxiran-2-yl)propyl]oxirane	9.802	52338-90-6	C10H16O3	0.27 ± 0.01	ND	ND	ND
VFC61	3,3-Dimethylhexane	16.34	563-16-6	C8H18	ND	ND	ND	0.29 ± 0.01
	Total				3.93 ± 0.15	5.55 ± 0.40	5.33 ± 0.29	1.53 ± 0.06
Pyrazines(2)								
VFC62	2,5-Dimethylpyrazine	5.5	123-32-0	C6H8N2	0.52 ± 0.03	0.96 ± 0.02	0.39 ± 0.02	ND
VFC63	2-Ethyl-3,5-dimethylpyrazine	10.13	13925-07-0	C8H12N2	ND	0.85 ± 0.04	0.49 ± 0.01	ND
	Total				0.52 ± 0.03	1.81 ± 0.06	0.88 ± 0.03	0
Furan(1)								
VFC64	2-Pentylfuran	7.61	3777-69-3	C9H14O	1.28 ± 0.12	2.40 ± 0.17	3.05 ± 0.11	2.37 ± 0.13
	Total				1.28 ± 0.12	2.40 ± 0.17	3.05 ± 0.11	2.37 ± 0.13

MS: Identification based on the NIST 14 (Department of commerce, USA) mass spectral database. a: The relative contents are the means of three repetitions ± standard deviation. b: OB, Oat bran; S-OB, Steamed oat bran; M-OB, Microwaved oat bran; HA-OB, Hot air dried oat bran. c: ND: Not detected.

**Table 3 foods-11-03070-t003:** Contents and composition of amino acids and in oat bran.

Amino Acids	Amino Acid Content (g/100 g)			
	OB	S-OB	M-OB	HA-OB
Sweet amino acids				
Ala	1.09 ± 0.01 ^a^	1.04 ± 0.01 ^a^	1.05 ± 0.01 ^a^	1.06 ± 0.04 ^a^
Gly	1.12 ± 0.01 ^a^	1.02 ± 0.01 ^b^	1.05 ± 0.02 ^ab^	1.04 ± 0.04 ^ab^
Ser	0.98 ± 0.01 ^a^	0.92 ± 0.01 ^a^	0.93 ± 0.02 ^a^	0.94 ± 0.03 ^a^
Thr *	0.71 ± 0.01 ^a^	0.68 ± 0.01 ^a^	0.69 ± 0.01 ^a^	0.70 ± 0.02 ^a^
Pro	1.12 ± 0.06 ^ab^	1.19 ± 0.04 ^a^	1.11 ± 0.02 ^ab^	1.03 ± 0.04 ^b^
Total	5.02 ± 0.10 ^a^	4.85 ± 0.08 ^a^	4.83 ± 0.08 ^a^	4.77 ± 0.17 ^a^
Umami amino acids				
Glu	4.00 ± 0.04 ^a^	3.87 ± 0.03 ^b^	3.85 ± 0.06 ^b^	3.75 ± 0.1 ^b^
Asp	1.63 ± 0.01 ^a^	1.55 ± 0.01 ^a^	1.56 ± 0.03 ^a^	1.59 ± 0.05 ^a^
Total	5.63 ± 0.05 ^a^	5.42 ± 0.04 ^a^	5.41 ± 0.09 ^a^	5.34 ± 0.15 ^a^
Bitter amino acids				
Val *	1.04 ± 0.01 ^a^	0.99 ± 0 ^a^	1.00 ± 0.22 ^a^	1.00 ± 0.04 ^a^
Met *	0.16 ± 0.05 ^ab^	0.07 ± 0.01 ^a^	0.21 ± 0.08 ^b^	0.04 ± 0.04 ^b^
Leu *	1.53 ± 0.02 ^a^	1.46 ± 0.01 ^ab^	1.45 ± 0.02 ^ab^	1.45 ± 0.05 ^b^
Ile *	0.65 ± 0.02 ^a^	0.60 ± 0 ^b^	0.61 ± 0 ^b^	0.62 ± 0.03 ^b^
Phe *	1 ± 0.01 ^a^	0.97 ± 0.03 ^a^	0.95 ± 0.01 ^a^	0.98 ± 0.04 ^a^
His	0.48 ± 0.03 ^a^	0.42 ± 0.01 ^b^	0.43 ± 0.01 ^b^	0.44 ± 0.01 ^b^
Arg *	1.22 ± 0.02 ^a^	1.15 ± 0.01 ^b^	1.20 ± 0.02 ^ab^	1.17 ± 0.05 ^b^
Total	6.08 ± 0.16 ^a^	5.66 ± 0.07 ^a^	5.85 ± 0.36 ^a^	5.70 ± 0.26 ^a^
Tasteless amino acids				
Lys *	0.82 ± 0.00 ^a^	0.75 ± 0.01 ^a^	0.79 ± 0.02 ^a^	0.78 ± 0.03 ^a^
Tyr	0.55 ± 0.03 ^a^	0.51 ± 0.01 ^a^	0.57 ± 0 ^a^	0.56 ± 0.03 ^a^
Total	1.37 ± 0.03 ^a^	1.26 ± 0.02 ^a^	1.36 ± 0.02 ^a^	1.34 ± 0.06 ^a^
Other amino acids				
Cys	0.56 ± 0.01 ^a^	0.52 ± 0.01 ^a^	0.52 ± 0.01 ^a^	0.25 ± 0.20 ^b^
Total	0.56 ± 0.01 ^a^	0.52 ± 0.01 ^a^	0.52 ± 0.01 ^a^	0.25 ± 0.20 ^b^
Essential amino acids	6.39 ± 0.15 ^a^	5.94 ± 0.08 ^a^	6.13 ± 0.37 ^a^	6.01 ± 0.26 ^a^
Sum of amino acids	18.66 ± 0.35 ^a^	17.71 ± 0.22 ^a^	17.97 ± 0.56 ^a^	17.40 ± 0.84 ^a^
Essential Amino Acids/Sum of amino acids	34.24 ± 0.16 ^a^	33.54 ± 0.04 ^b^	34.08 ± 0.10 ^a^	34.55 ± 0.17 ^a^

Mean ± SD (*n* = 3). Means with different superscript (a,b) in the same row are significantly different (*p* < 0.05).* indicates essential amino acids. OB, Oat bran; S-OB, Steamed oat bran; M-OB, Microwaved oat bran; HA-OB, Hot air dried oat bran.

**Table 4 foods-11-03070-t004:** Composition and content of fatty acids in oat bran after different pretreatments.

Fatty Acid Composition	Fatty Acid Composition (% of Total Fatty Acids)			
	OB	S-OB	M-OB	HA-OB
Palmitic acid (C16:0)	17.28 ± 0.04 ^a^	16.96 ± 0.05 ^b^	17.38 ± 0.08 ^a^	17.24 ± 0.02 ^a^
Stearic acid (C18:0)	1.60 ± 0.01 ^b^	1.60 ± 0.01 ^b^	1.65 ± 0.01 ^a^	1.63 ± 0.01 ^ab^
Saturated fatty acid	18.88 ± 0.05 ^a^	18.56 ± 0.06 ^b^	19.03 ± 0.08 ^a^	18.87 ± 0.03 ^a^
Oleic acid (C18:1)	46.98 ± 0.15 ^a^	45.97 ± 0.04 ^b^	46.49 ± 0.15 ^a^	46.58 ± 0.13 ^a^
Linoleic acid (C18:2)	31.38 ± 0.14 ^b^	32.69 ± 0.11 ^a^	31.65 ± 0.07 ^b^	31.76 ± 0.16 ^b^
linolenic acids (C18:3)	0.96 ± 0.01 ^a^	0.94 ± 0.00 ^a^	0.96 ± 0.01 ^a^	0.96 ± 0.01 ^a^
Unsaturated fatty acid	79.32 ± 0.31 ^b^	79.60 ± 0.16 ^a^	79.11 ± 0.22 ^b^	79.30 ± 0.30 ^b^

Mean ± SD (*n* = 3). Means with different superscript (a,b) in the same row are significantly different (*p* < 0.05).OB, Oat bran; S-OB, Steamed oat bran; M-OB, Microwaved oat bran; HA-OB, Hot air dried oat bran.

## Data Availability

The data presented in this study are available on request from the corresponding author.

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
