# Peer review of "Effects of Pretreatment on the Volatile Composition, Amino Acid, and Fatty Acid Content of Oat Bran"

_foods, 2022, doi:10.3390/foods11193070_

Round 1
Reviewer 1 Report
1. Title does not reflect fully what was conducted in the study (along with HS-SPME–2 GC-MS, amino acid and fatty acid content was also analysed). Title should concisely cover all segments of the manuscript and therefore, must be revised (complemented).
2. Abstract is not well conceptualized. Numbers in brackets specifying specific manuscript segments ((1) for Introduction, (2) for methods, (3) for results, etc.) should be removed. Other numbers indicating references (I suppose) in brackets (42, 25 36, etc.) should also be omitted (Abstract should not contain references). Revise/rewrite corresponding sentences in full form and indicate crucial HS-SPME–2 GC-MS results.
3. Keywords: “volatile composition” instead of “flavour compounds” (be consistent in terminology used throughout the entire manuscript).
4. Methods: I suggest using simple “analysis” instead of “assay” in subsection 2.3.4. (assay indicate a new and/or more complex procedure compared to conventional and routine analysis - I believe assay was not conducted here).
The same goes for section 2.3.5. on fatty acids. I see no references in section 2.3.5. - is this a new analytical approach? I believe some segments were published before/elsewhere. Indicate an appropriate reference for that and clearly state which protocol segments were modified / new here compared to previous analytical approach.
Statistical analysis: Software package used for PCA analysis should be clearly specified – name, version, city, state (the link on Chinese web page is not sufficient or appropriate reference).
The same goes for correlation / Spearman analysis – indicate software package used.
5. The results of PCA and correlation should be interpreted in more details. Figure 1 interpretation rather scarce and vague / unspecific, i.e. to general and non-informative.
Multivariate discriminant analyses, exploratory analyses, and chemometrics (pattern recognition methods in general) are much more than just presenting the outcome - visual clustering of particular samples based on particular analytical parameter. If there is no proper and/or sufficient description of modelling, as well as proper/sufficient interpretation and elaboration of the results obtained, such presentation is defined as scientifically questionable. Therefore, it is necessary to incorporate more detailed info for PCA applied - info about input data, latent variables, what exactly was captured in eigenvalues compressed in PC1 and PC2, etc. Methodology section and Results and discussion section should be complemented in this regard. In addition, the title of Figure 1 is not appropriate - the figure is not representing “analysis” but result (“The analysis of PCA by electronic nose…” is incorrect figure title) – revise as “PCA scatter plot of…”
Figure captions should be concise but informative enough and self-explanatory.
6. Results: Table 2 should be completely revised. There is need to use IUPAC nomenclature for the compounds in Table 2, such as: 3-Methylbutanal, (E)-Hex-2-enal, (E,E)-Nona-2,4-dienal, Vinyl caproate, Heptan-1-ol, others.. Those are just a few examples, but EACH COMPOUND name should be rechecked and written correctly according to IUPAC rules.
7. The last sentence of Conclusion is also rather vague and non-specific. The study has demonstrated more than “This study provides technical reference and taste evaluation for oat bran processing.”. Moreover, to what is “technical reference” mentioned here actually referring to? Also, a term “taste evaluation” is rather colloquial and not entirely consistent with what presented in the study. Authors should overview concisely overall results obtained in this study, and indicate potential investigations for future research on this matter. Please, revise the last part of Conclusion accordingly.
Author Response
Please see the attachment。

Reviewer 2 Report
Dear Authors.
Below you will find the observations, comments and suggested changes to your manuscript.
1) From the abstract delete numbers (1), (2), (3) and (4).
2) Line 71-72: Make a correction to the reference.
3) En 2.1 - Include the purity value and concentration of the substances used in the study.
4) The references are not easy to read and analyse. Diagram the references again.
5) Lines 91-92: Did the grinding of the sample generate heat ? Yes or not? If there is heat can it significantly generate artifactual compounds?
6) Lines 95-100: Can you include the temperature value for each completed process?
7) Lines 130-131: Indicate the operating conditions of the GC/MS equipment, such as injection port temperature, oven heating ramp, column characteristics (manufacturer, reference, series, dimensions, etc.), as well as the MS.
For example, a model of how it is written is:
Analyses were performed in a gas chromatograph Agilent 6850 Series II Network System equipped with mass selective detector Agilent 5975B VL (Electron impact ionization, EI, 70 eV), a split/splitless injector (1:100 split ratio) and Enhanced ChemStation MSD D.03.00.52 data system, that included the spectral libraries Wiley and Nist. A fused-silica capillary column is a HP-1MS (30 m x 0.20 mm i.d, 0.33 μm film thickness) was used. Chromatographic conditions were: The GC oven temperature was programmed from 100 °C (2 min) to 285 °C (10 min) at 25 °C/min slope and post run to 320 °C (3 min). The temperatures the injection port, ionization chamber and the transfer line were set at 300, 185 and 285 °C, respectively. Helium (99.999%, AGA-Fano) was used as carrier gas, with 85 kPa column head pressure and linear velocity at constant flow (1 mL/min). Mass spectra and reconstructed (total) ion chromatograms were obtained by scanning in the mass range m/z 30-500 Da at 2.2 scan/s. Chromatographic peaks were checked for homogeneity with the aid of the mass chromatograms for the characteristic fragment ions and with the help of the peak purity program. Tentative identification criteria of compounds were based on coincidence percentage (≥ 85%) of obtained compounds compared to Wiley 7n.1 and Nist 05a.L mass spectra libraries.
It is similar to what is described in the lines 161-166 (where they indicate the operating conditions of the GC/FID system).
8) Line 135: Hydrochloric acid is denoted as HCl no HCL.
9) Line 157: Stratification or esterification?
10) Line 160: Did you inject the CG with 0.8 mL (800 uL!!) and in splitless mode? Are you sure?
11) Line 168: What is the value of n=?
12) The electronic nose is a device capable of comparing and classifying the aromas of food products, usually qualitatively. Its main applications in quality control are: - process monitoring, determination of defects or contaminants, freshness assessment and classification of freshness; - freshness assessment and classification of samples according to origin or variety. Then (lines 190-192) how does flavour correlate with the aromas of thermal processes?
13) The electronic nose will not be able to evaluate a food for which it is not programmed. For calibration it is necessary to have already characterised standard samples that have been previously evaluated by a panel of experts. Then, how did you calibrate the electronic nose?
14) Can you indicate whether acrylamide and glycidamide were formed during the thermal processing of the samples? Have you thought about determining acrylamide levels?
15) Redesign table 2, it is not easy to read or interpret. What is the value of n=?. I do not agree with the expression "Flavor substance", in my concept it is "volatile compound". Include a column in the table and indicate the match value with the spectra in the MS database. Also write down which database you used for the comparison (name, reference, manufacturer, etc).
16) In table 3, some amino acids are marked with an asterisk, why are there no notations or notes in the table? Statistical differences should be reported in superscript format. Redesign the table.
17) In the section 3.5.1. - Because the content of saturated fatty acids increases or decreases in unsaturated fatty acids, what is the cause? What are the variables/causes affecting lipid peroxidation?
18) At the end of the discussion include a paragraph on the strengths and weaknesses of your work.
19) In my opinion the paper has a conceptual and analytical deficiency: The samples of the three thermal processes were not compared with an analytical control like the sample without thermal process (control sample), therefore it is not possible to indicate/interpret which process is better on taste and aroma properties.
Please update the manuscript.
Regards,
Reviewer
Round 2
Reviewer 1 Report
Dear Authors,
You have revised the manuscript in accordance with provided suggestions / comments / guidelines for revision. The revision was carried out thoroughly and carefully by taking into consideration each and every question raised, which is highly appreciated (commendation to Xue Bai and corresponding author for this).
I have only two more suggestions for the authors:
1. to check if any of compounds listed in Table 2 is potentially an artefact and/or contamination; this should always be considered in GC/MS-based analyses (if any detected - these compounds should be removed from Table 2).
2. to revise the title as follows: “Effects of pretreatment on the volatile composition, amino acid, and fatty acid content of oat bran” (there is no need to accentuate correlation analysis in the title).
Author Response
Thank you for your careful review and constructive suggestions regarding our manuscript. We have revised the manuscript in accordance with the reviewer’s comments.Please see the attachment.

Reviewer 2 Report
Dear Authors.
Thaks for your answers. Please make the suggested changes.
1) Lines 104-105 - Include the temperature that was generated during the process. e.g. Microwaving oat brans were microwaved for 2 min at 800 W. The completed process temperature was close to 100 °C
2). Line 154 - Include the value of the minimum match percentage taken as a baseline.
I look forward to hearing from you soon.
Best regards
Reviewer 2
Author Response

(The authors gave the same response as above.)
